# Large Language Models are Zero-Shot Reasoners

**Takeshi Kojima**
The University of Tokyo
t.kojima@weblab.t.u-tokyo.ac.jp

**Shixiang Shane Gu**
Google Research, Brain Team

**Machel Reid**
Google Research[*]

**Yutaka Matsuo**
The University of Tokyo

**Yusuke Iwasawa**
The University of Tokyo

## Abstract

Pretrained large language models (LLMs) are widely used in many sub-fields of natural language processing (NLP) and generally known as excellent *few-shot* learners with task-specific exemplars. Notably, chain of thought (CoT) prompting, a recent technique for eliciting complex multi-step reasoning through step-by-step answer examples, achieved the state-of-the-art performances in arithmetics and symbolic reasoning, difficult *system-2* tasks that do not follow the standard scaling laws for LLMs. While these successes are often attributed to LLMs' ability for few-shot learning, we show that LLMs are decent *zero-shot* reasoners by simply adding "Let's think step by step" before each answer. Experimental results demonstrate that our Zero-shot-CoT, using the same single prompt template, significantly outperforms zero-shot LLM performances on diverse benchmark reasoning tasks including arithmetics (MultiArith, GSM8K, AQUA-RAT, SVAMP), symbolic reasoning (Last Letter, Coin Flip), and other logical reasoning tasks (Date Understanding, Tracking Shuffled Objects), without any hand-crafted few-shot examples, e.g. increasing the accuracy on MultiArith from 17.7% to 78.7% and GSM8K from 10.4% to 40.7% with large-scale InstructGPT model (text-davinci-002), as well as similar magnitudes of improvements with another off-the-shelf large model, 540B parameter PaLM. The versatility of this single prompt across very diverse reasoning tasks hints at untapped and understudied fundamental *zero-shot* capabilities of LLMs, suggesting high-level, multi-task broad cognitive capabilities may be extracted by simple prompting. We hope our work not only serves as the minimal strongest zero-shot baseline for the challenging reasoning benchmarks, but also highlights the importance of carefully exploring and analyzing the enormous zero-shot knowledge hidden inside LLMs before crafting finetuning datasets or few-shot exemplars.

## 1   Introduction

Scaling up the size of language models has been key ingredients of recent revolutions in natural language processing (NLP) [Vaswani et al., 2017, Devlin et al., 2019, Raffel et al., 2020, Brown et al., 2020, Thoppilan et al., 2022, Rae et al., 2021, Chowdhery et al., 2022]. The success of large language models (LLMs) is often attributed to (in-context) few-shot or zero-shot learning. It can solve various tasks by simply conditioning the models on a few examples (few-shot) or instructions describing the task (zero-shot). The method of conditioning the language model is called "prompting" [Liu et al., 2021b], and designing prompts either manually [Schick and Schütze, 2021, Reynolds and McDonell, 2021] or automatically [Gao et al., 2021, Shin et al., 2020] has become a hot topic in NLP.

---

[*]Work done while at The University of Tokyo.

36th Conference on Neural Information Processing Systems (NeurIPS 2022).

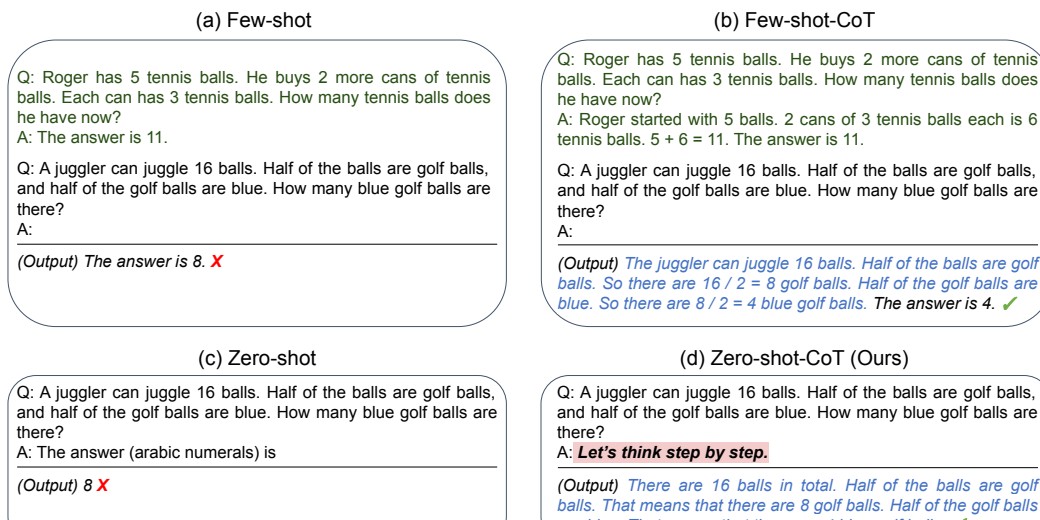

Figure 1: Example inputs and outputs of GPT-3 with (a) standard Few-shot ([Brown et al., 2020]), (b) Few-shot-CoT ([Wei et al., 2022]), (c) standard Zero-shot, and (d) ours (Zero-shot-CoT). Similar to Few-shot-CoT, Zero-shot-CoT facilitates multi-step reasoning (blue text) and reach correct answer where standard prompting fails. Unlike Few-shot-CoT using step-by-step reasoning examples **per task**, ours does not need any examples and just uses the same prompt "Let's think step by step" *across all tasks* (arithmetic, symbolic, commonsense, and other logical reasoning tasks).

In contrast to the excellent performance of LLMs in intuitive and single-step *system-1* [Stanovich and West, 2000] tasks with task-specific few-shot or zero-shot prompting [Liu et al., 2021b], even language models at the scale of 100B or more parameters had struggled on *system-2* tasks requiring slow and multi-step reasoning [Rae et al., 2021]. To address this shortcoming, Wei et al. [2022], Wang et al. [2022] have proposed *chain of thought* prompting (CoT), which feed LLMs with the step-by-step reasoning examples rather than standard question and answer examples (see Fig. 1-a). Such chain of thought demonstrations facilitate models to generate a reasoning path that decomposes the complex reasoning into multiple easier steps. Notably with CoT, the reasoning performance then satisfies the scaling laws better and jumps up with the size of the language models. For example, when combined with the 540B parameter PaLM model [Chowdhery et al., 2022], chain of thought prompting significantly increases the performance over standard few-shot prompting across several benchmark reasoning tasks, e.g., GSM8K (17.9% → 58.1%).

While the successes of CoT prompting [Wei et al., 2022], along those of many other task-specific prompting work [Gao et al., 2021, Schick and Schütze, 2021, Liu et al., 2021b], are often attributed to LLMs' ability for few-shot learning [Brown et al., 2020], we show that LLMs are decent *zero-shot* reasoners by adding a simple prompt, *Let's think step by step*, to facilitate step-by-step thinking before answering each question (see Figure 1). Despite the simplicity, our Zero-shot-CoT successfully generates a plausible reasoning path in a zero-shot manner and reaches the correct answer in a problem where the standard zero-shot approach fails. Importantly, our Zero-shot-CoT is versatile and *task-agnostic*, unlike most prior task-specific prompt engineering in the forms of examples (few-shot) or templates (zero-shot) [Liu et al., 2021b]: it can facilitate step-by-step answers across various reasoning tasks, including arithmetic (MultiArith [Roy and Roth, 2015], GSM8K [Cobbe et al., 2021], AQUA-RAT [Ling et al., 2017], and SVAMP [Patel et al., 2021]), symbolic reasoning (Last letter and Coin flip), commonsense reasoning (CommonSenseQA [Talmor et al., 2019] and Strategy QA [Geva et al., 2021]), and other logical reasoning tasks (Date understanding and Tracking Shuffled Objects from BIG-bench [Srivastava et al., 2022]) without modifying the prompt per task.

We empirically evaluate Zero-shot-CoT against other prompting baselines in Table 2. While our Zero-shot-CoT underperforms Few-shot-CoT with carefully-crafted and task-specific step-by-step examples, Zero-shot-CoT achieves enormous score gains compared to the zero-shot baseline, e.g. from 17.7% to 78.7% on MultiArith and from 10.4% to 40.7% on GSM8K with large-scale InstructGPT

model (text-davinci-002). We also evaluate Zero-shot-CoT with another off-the-shelf large model, 540B parameter PaLM, showing similar magnitudes of improvements on MultiArith and GSM8K. Importantly, with our single fixed prompt, zero-shot LLMs have a significantly better scaling curve comparable to that of the few-shot CoT baseline. We also show that besides Few-shot-CoT requiring human engineering of multi-step reasoning prompts, their performance deteriorates if prompt example question types and task question type are unmatched, suggesting high sensitivity to per-task prompt designs. In contrast, the versatility of this single prompt across diverse reasoning tasks hints at untapped and understudied *zero-shot* fundamental capabilities of LLMs, such as higher-level broad cognitive capabilities like generic logical reasoning [Chollet, 2019]. While the vibrant field of LLMs started out from the premise of excellent few-shot learners [Brown et al., 2020], we hope our work encourages more research into uncovering *high-level* and *multi-task* zero-shot capabilities hidden inside those models.

## 2 Background

We briefly review the two core preliminary concepts that form the basis of this work: the advent of large language models (LLMs) and prompting, and chain of thought (CoT) prompting for multi-step reasoning.

**Large language models and prompting** A language model (LM), is a model that looks to estimate the probability distribution over text. Recently, scaling improvements through larger model sizes (from a few million [Merity et al., 2016] to hundreds of millions [Devlin et al., 2019] to hundreds of billions [Brown et al., 2020] parameters) and larger data (e.g. webtext corpora [Gao et al., 2020]) have enabled pre-trained large language models (LLMs) to be incredibly adept at many downstream NLP tasks. Besides the classic "pre-train and fine-tune" paradigm [Liu et al., 2021b], models scaled to 100B+ parameters exhibit properties conducive to few-shot learning [Brown et al., 2020], by way of in context learning, where one can use a text or template known as a *prompt* to strongly guide the generation to output answers for desired tasks, thus beginning an era of "pre-train and prompt" [Liu et al., 2021a]. In work, we call such prompts with explicit conditioning on few task examples as *few-shot* prompts, and other template-only prompts as *zero-shot* prompts.

**Chain of thought prompting** Multi-step arithmetic and logical reasoning benchmarks have particularly challenged the scaling laws of large language models [Rae et al., 2021]. Chain of thought (CoT) prompting [Wei et al., 2022], an instance of few-shot prompting, proposed a simple solution by modifying the answers in few-shot examples to step-by-step answers, and achieved significant boosts in performance across these difficult benchmarks, especially when combined with very large language models like PaLM [Chowdhery et al., 2022]. The top row of Figure 1 shows standard few-shot prompting against (few-shot) CoT prompting. Notably, few-shot learning was taken as a given for tackling such difficult tasks, and the zero-shot baseline performances were not even reported in the original work [Wei et al., 2022]. To differentiate it from our method, we call Wei et al. [2022] as *Few-shot-CoT* in this work.

## 3 Zero-shot Chain of Thought

We propose Zero-shot-CoT, a zero-shot template-based prompting for chain of thought reasoning. It differs from the original chain of thought prompting [Wei et al., 2022] as it does not require step-by-step few-shot examples, and it differs from most of the prior template prompting [Liu et al., 2021b] as it is inherently task-agnostic and elicits multi-hop reasoning across a wide range of tasks with a single template. The core idea of our method is simple, as described in Figure 1: add *Let's think step by step*, or a a similar text (see Table 4), to extract step-by-step reasoning.

### 3.1 Two-stage prompting

While Zero-shot-CoT is conceptually simple, it uses prompting twice to extract both reasoning and answer, as explained in Figure 2. In contrast, the zero-shot baseline (see the bottom-left in Figure 1) already uses prompting in the form of "The answer is", to extract the answers in correct formats. Few-shot prompting, standard or CoT, avoids needing such answer-extraction prompting by explicitly designing the few-shot example answers to end in such formats (see the top-right and top-left

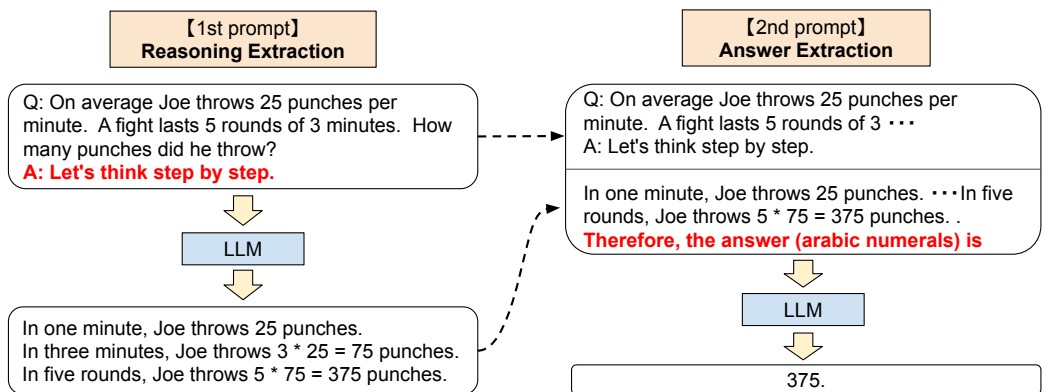

Figure 2: Full pipeline of Zero-shot-CoT as described in § 3: we first use the first "reasoning" prompt to extract a full reasoning path from a language model, and then use the second "answer" prompt to extract the answer in the correct format from the reasoning text.

in Figure 1). In summary, Few-shot-CoT [Wei et al., 2022] requires careful human engineering of a few prompt examples with specific answer formats per task, while Zero-shot-CoT requires less engineering but requires prompting LLMs twice.

**1st prompt: reasoning extraction**   In this step we first modify the input question $\mathbf{x}$ into a *prompt* $\mathbf{x}'$ using a simple template "Q: [X]. A: [T]", where [X] is an input slot for $\mathbf{x}$ and [T] is an slot for hand-crafted trigger sentence $\mathbf{t}$ that would extract chain of though to answer the question $\mathbf{x}$. For example, if we use "Let's think step by step" as a trigger sentence, the prompt $\mathbf{x}'$ would be "Q: [X]. A: Let's think step by step.". See Table 4 for more trigger examples. Prompted text $\mathbf{x}'$ is then fed into a language model and generate subsequent sentence $\mathbf{z}$. We can use any decoding strategy, but we used greedy decoding throughout the paper for the simplicity.

**2nd prompt: answer extraction**   In the second step, we use generated sentence $\mathbf{z}$ along with prompted sentence $\mathbf{x}'$ to extract the final answer from the language model. To be concrete, we simply concatenate three elements as with "[X'] [Z] [A]": [X'] for 1st prompt $\mathbf{x}'$, [Z] for sentence $\mathbf{z}$ generated at the first step, and [A] for a trigger sentence to extract answer. The prompt for this step is *self-augmented*, since the prompt contains the sentence $\mathbf{z}$ generated by the same language model. In experiment, we use slightly different answer trigger depending on the answer format. For example, we use "Therefore, among A through E, the answer is" for multi-choice QA, and "Therefore, the answer (arabic numerals) is" for math problem requiring numerical answer. See Appendix A.5 for the lists of answer trigger sentences. Finally, the language model is fed the prompted text as input to generate sentences $\hat{\mathbf{y}}$ and parse the final answer. See "Answer Cleansing" at §4 for the parser details.

## 4   Experiment

**Tasks and datasets**   We evaluate our proposal on 12 datasets from four categories of reasoning tasks: arithmetic, commonsense, symbolic, and other logical reasoning tasks. See Appendix A.2 for the detailed description of each datasets.

For arithmetic reasoning, we consider the following six datasets: (1) SingleEq [Koncel-Kedziorski et al., 2015], (2) AddSub [Hosseini et al., 2014], (3) MultiArith [Roy and Roth, 2015], (4) AQUA-RAT [Ling et al., 2017], (5) GSM8K [Cobbe et al., 2021], and (6) SVAMP [Patel et al., 2021]. The first three are from the classic Math World Problem Repository [Koncel-Kedziorski et al., 2016], and the last three are from more recent benchmarks. SingleEq and AddSub contain easier problems, which do not require multi-step calculation to solve the tasks. MultiArith, AQUA-RAT, GSM8k, and SVAMP are more challenging datasets that require multi-step reasoning to solve.

For commonsense reasoning, we use CommonsenseQA [Talmor et al., 2019] and StrategyQA [Geva et al., 2021]. CommonsenseQA asks questions with complex semantics that often require reasoning

based on prior knowledge [Talmor et al., 2019]. StrategyQA requires models to infer an implicit multi-hop reasoning to answer questions [Geva et al., 2021].

For symbolic reasoning, we use Last Letter Concatenation and Coin Flip [Wei et al., 2022]. Last letter Concatenation asks the model to concatenate the last letters of each word. We used randomly selected four names for each sample. Coin Flip asks the model to answer whether a coin is still heads up after people either flip or do not flip the coin. We created samples of four times flip or not flip trials. Although these tasks are easy for humans, LMs typically exhibit a flat scaling curve.

For other logical reasoning tasks, we choose two evaluation sets from the BIG-bench effort [Srivastava et al., 2022]: Date Understanding [2] and Tracking Shuffled Objects. Date Understanding asks models to infer the date from a context. Tracking Shuffled Objects tests a model's ability to infer the final state of objects given its initial state and a sequence of object shuffling. We used a dataset of tracking three shuffled objects for our experiment.

**Models** We experiment with 17 models in total. Main experiments are conducted with Instruct-GPT3 [Ouyang et al., 2022] (text-ada/babbage/curie/davinci-001 and text-davinci-002)[3], original GPT3 [Brown et al., 2020] (ada, babbage, curie, and davinci)[4], and PaLM [Chowdhery et al., 2022] (8B, 62B, and 540B). In addition, we used GPT-2[Radford et al., 2019], GPT-Neo[Black et al., 2021], GPT-J[Wang and Komatsuzaki, 2021], T0 [Sanh et al., 2022], and OPT [Zhang et al., 2022] for model scaling study. The size of LMs ranges from 0.3B to 540B. We include both standard (e.g. GPT-3 and OPT), and instruction following variants (e.g. Instruct-GPT3 and T0). See Appendix A.3 for model description details. Unless otherwise stated, we use text-davinci-002 throughout the experiments.

**Baselines** We compare our Zero-shot-CoT mainly to standard Zero-shot prompting to verify the effectiveness of its chain of thought reasoning. For Zero-shot experiments, similar answer prompts as Zero-shot-CoT are used as default. See Appendix A.5 for detail. To better evaluate the zero-shot ability of LLMs on reasoning tasks, we also compare our method to Few-shot and Few-shot-CoT baselines from [Wei et al., 2022], using the same in-context examples. Throughout the experiments, we use greedy decoding across all the methods. For the zero-shot approaches, the results are therefore deterministic. For the few-shot approaches, since the order of in-context examples could affect the results [Lu et al., 2022], we run each experiment only once with a fixed seed across all methods and datasets, for fair comparisons with the zero-shot methods. Wei et al. [2022] showed that the order of examples did not cause large variance in CoT experiments.

**Answer cleansing** After the model outputs a text by answer extraction (see § 3 and Figure 2), our method picks up only the part of the answer text that first satisfies the answer format. For example, if the answer prompting outputs "probably 375 and 376" on arithmetic tasks, we extract the first number "375" and set it as the model prediction. In the case of multiple-choice, the first large letter we encounter is set as the prediction. See Appendix A.6 for more detail. Standard Zero-shot method follows the same idea. For Few-shot and Few-shot-CoT methods, we follow [Wang et al., 2022] and first extract the answer text after "The answer is " from the model output, and apply the same answer cleansing to parse the answer text. If "The answer is" is not found in the model output, we search from the back of the text and set the first text that satisfies the answer format as the prediction.

## 4.1 Results

**Zero-shot-CoT vs. Zero-shot** Table 1 summarize accuracy of our method (Zero-shot-CoT) and standard zero-shot prompting (Zero-shot) for each dataset. Zero-shot-CoT substantially outperforms four out of six arithmetic reasoning tasks (MultiArith, GSM8K, AQUA, SVAMP), all symbolic reasoning, and all other logical reasoning tasks (from BIG-bench [Srivastava et al., 2022]). For

---

[2]While prior work [Wei et al., 2022] categorized Date Understanding task into Common Sense reasoning, our study categorized this task into logical reasoning because this task requires less prior knowledge and more logical reasoning between dates.

[3]Our experiment for Instruct GPT-3 models includes both text-****-001 and text-davinci-002. Text-davinci-002 differs from text-****-001 in that they use different fine-tuning data depending on the date range collected from the APIs. Specifically, text-davinci-002 uses data up to Jun 2021, while text-****-001 uses data up to Oct 2019. (See `https://beta.openai.com/docs/engines/gpt-3`)

[4]Our experiments with GPT3 series are conducted by using OpenAI API between April-2022 and May-2022, except for No.10-16 in Table 4 in Aug-2022.

Table 1: Accuracy comparison of Zero-shot-CoT with Zero-shot on each tasks. The values on the left side of each task are the results of using answer extraction prompts depending on answer format as described at § 3. The values on the right side are the result of additional experiment where standard answer prompt "The answer is" is used for answer extraction. See Appendix A.5 for detail setups.

| | Arithmetic | | | | | |
|---|---|---|---|---|---|---|
| | SingleEq | AddSub | MultiArith | GSM8K | AQUA | SVAMP |
| zero-shot | 74.6/**78.7** | **72.2/77.0** | 17.7/22.7 | 10.4/12.5 | 22.4/22.4 | 58.8/58.7 |
| zero-shot-cot | **78.0**/78.7 | 69.6/74.7 | **78.7/79.3** | **40.7**/40.5 | **33.5**/31.9 | **62.1/63.7** |

| | Common Sense | | Other Reasoning Tasks | | Symbolic Reasoning | |
|---|---|---|---|---|---|---|
| | Common SenseQA | Strategy QA | Date Understand | Shuffled Objects | Last Letter (4 words) | Coin Flip (4 times) |
| zero-shot | **68.8/72.6** | 12.7/**54.3** | 49.3/33.6 | 31.3/29.7 | 0.2/- | 12.8/53.8 |
| zero-shot-cot | 64.6/64.0 | **54.8**/52.3 | **67.5/61.8** | **52.4/52.9** | **57.6**/- | **91.4/87.8** |

Table 2: Comparison with baseline methods using accuracies on MultiArith and GSM8K. text-davinci-002 is used as the model if not specified. We used the same 8 examples as described in [Wei et al., 2022] for Few-shot and Few-shot-CoT settings. (*1) To verify the variance of changing examples, we report two results for 4-shot-cot by splitting the eight examples into two groups. (*2) We insert "Let's think step by step." at the beginning of answer part of each exemplars for Few-shot-CoT to test performance gains. Further experiment results with PaLM are found at Appendix D

| | MultiArith | GSM8K |
|---|---|---|
| **Zero-Shot** | **17.7** | **10.4** |
| Few-Shot (2 samples) | 33.7 | 15.6 |
| Few-Shot (8 samples) | 33.8 | 15.6 |
| **Zero-Shot-CoT** | **78.7** | **40.7** |
| Few-Shot-CoT (2 samples) | 84.8 | 41.3 |
| Few-Shot-CoT (4 samples : First) (*1) | 89.2 | - |
| Few-Shot-CoT (4 samples : Second) (*1) | 90.5 | - |
| Few-Shot-CoT (8 samples) | 93.0 | 48.7 |
| **Zero-Plus-Few-Shot-CoT (8 samples)** (*2) | **92.8** | **51.5** |
| Finetuned GPT-3 175B [Wei et al., 2022] | - | 33 |
| Finetuned GPT-3 175B + verifier [Wei et al., 2022] | - | 55 |
| **PaLM 540B: Zero-Shot** | **25.5** | **12.5** |
| **PaLM 540B: Zero-Shot-CoT** | **66.1** | **43.0** |
| **PaLM 540B: Zero-Shot-CoT + self consistency** | **89.0** | **70.1** |
| PaLM 540B: Few-Shot [Wei et al., 2022] | - | 17.9 |
| PaLM 540B: Few-Shot-CoT [Wei et al., 2022] | - | 56.9 |
| PaLM 540B: Few-Shot-CoT + self consistency [Wang et al., 2022] | - | 74.4 |

example, Zero-shot-CoT achieves score gains from 17.7% to 78.7% on MultiArith and from 10.4% to 40.7% on GSM8K. Our method gives on-par performances for the remaining two arithmetic reasoning tasks (SingleEq and AddSub), which is expected since they do not require multi-step reasoning.

In commonsense reasoning tasks, Zero-shot-CoT does not provide performance gains. It is expected as Wei et al. [2022] also reports that even Few-shot-CoT does not provide performance gains on Lambda (135B), but does improve StrategyQA when combined with substantially larger PaLM (540B) model, which may also apply for ours. More importantly, we observe that many generated chain of thought themselves are surprisingly *logically* correct or only contains human-understandable mistakes (See Table 3), suggesting that Zero-shot-CoT does elicit for better commonsense reasoning even when the task metrics do not directly reflect it. We provide samples generated by Zero-shot-CoT for each dataset in Appendix B.

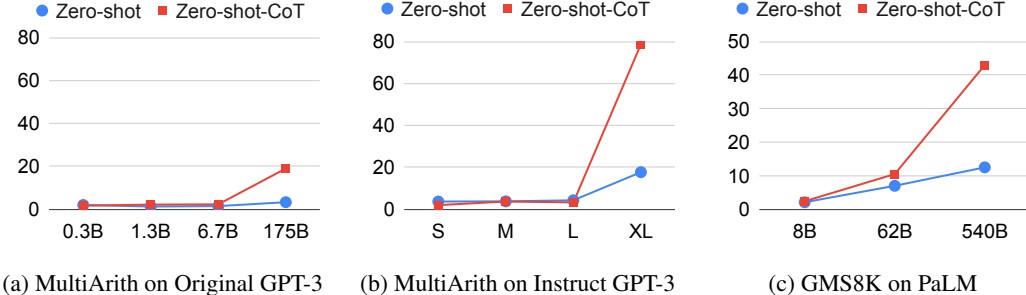

(a) MultiArith on Original GPT-3     (b) MultiArith on Instruct GPT-3     (c) GMS8K on PaLM

Figure 3: Model scale study with various types of models. S: text-ada-001, M: text-babbage-001, L: text-curie-001, XL: text-davinci-002. See Appendix A.3 and E for the detail.

Table 3: Examples generated by Zero-Shot-CoT on CommonsenseQA for Error Analysis.

| Example 1 **(Reasoning is correct.)** | Example 2 **(Model cannot choose only one.)** |
|---|---|
| **Q:** Where is a well used toy car likely to be found? Answer Choices: (A) child's room (B) boy's bedroom (C) own home (D) toy store (E) house | **Q:** What would be necessary for getting in shape? Answer Choices: (A) good health (B) exercise (C) muscle tone (D) sweat (E) feel better |
| **A: Let's think step by step.** A toy car is likely to be found in a child's room. **A child's room is likely to be found in a house.** Therefore, a toy car is likely to be found in a house. **Therefore, among A through E, the answer is** E. ✗ **(GT : A, Zero-shot: A)** | **A: Let's think step by step.** In order to get in shape, you need to be able to do three things: 1. Exercise regularly 2. Eat a healthy diet 3. Get enough sleep If you can do all three of those things, then you will be well on your way to getting in shape! **Therefore, among A through E, the answer is B, C, and D. (GT : B, Zero-shot: B)** |

**Comparison with other baselines** Table 2 compares the performances on two arithmetic reasoning benchmarks (MultiArith and GSM8K) across Zero-shot-CoT and baselines. The large gap between standard prompting (1st block) and chain of thought prompting (2nd block) suggests that these tasks are difficult without eliciting multi-step reasoning. Major improvements are confirmed on both Instruct GPT-3 (text-davinci-002) and PaLM (540B) models (4th block). While Zero-shot-CoT naturally underperforms Few-shot-CoT, it substantially outperforms standard Few-shot prompting with even 8 examples per task. For GSM8K, Zero-shot-CoT with Instruct GPT-3 (text-davinci-002) also outperforms finetuned GPT-3 and standard few-shot prompting with large models (PaLM, 540B), reported in Wei et al. [2022] (3rd and 4th block). See App. D for more experiment results with PaLM.

**Does model size matter for zero-shot reasoning?** Figure 3 compares performance of various language models on MultiArith / GSM8K. Without chain of thought reasoning, the performance does not increase or increases slowly as the model scale is increased, i.e., the curve is mostly flat. In contrast, the performance drastically increases with chain of thought reasoning, as the model size gets bigger, for Original/Instruct GPT-3 and PaLM. When the model size is smaller, chain of thought reasoning is not effective. This result aligns with the few-shot experiment results in Wei et al. [2022]. Appendix E shows extensive experiment results using wider variety of language models, including GPT-2, GPT-Neo, GPT-J, T0, and OPT. We also manually investigated the quality of generated chain of thought, and large-scale models clearly demonstrate better reasoning (See Appendix B for the sampled outputs for each model).

**Error Analysis** To better understand the behavior of Zero-shot-CoT, we manually investigated randomly selected examples generated by Instruct-GPT3 with Zero-shot-CoT prompting. See Appendix C for examples, where some of the observations include: (1) In commonsense reasoning (CommonsenseQA), Zero-shot-CoT often produces flexible and reasonable chain of thought even when the final prediction is not correct. Zero-shot-CoT often output multiple answer choices when the model find it is difficult to narrow it down to one (see Table 3 for examples). (2) In arithmetic

Table 4: Robustness study against template measured on the MultiArith dataset with text-davinci-002. (*1) This template is used in Ahn et al. [2022] where a language model is prompted to generate step-by-step actions given a high-level instruction for controlling robotic actions. (*2) This template is used in Reynolds and McDonell [2021] but is not quantitatively evaluated.

| No. | Category | Template | Accuracy |
|---|---|---|---|
| 1 | instructive | Let's think step by step. | **78.7** |
| 2 | | First, (*1) | 77.3 |
| 3 | | Let's think about this logically. | 74.5 |
| 4 | | Let's solve this problem by splitting it into steps. (*2) | 72.2 |
| 5 | | Let's be realistic and think step by step. | 70.8 |
| 6 | | Let's think like a detective step by step. | 70.3 |
| 7 | | Let's think | 57.5 |
| 8 | | Before we dive into the answer, | 55.7 |
| 9 | | The answer is after the proof. | 45.7 |
| 10 | misleading | Don't think. Just feel. | 18.8 |
| 11 | | Let's think step by step but reach an incorrect answer. | 18.7 |
| 12 | | Let's count the number of "a" in the question. | 16.7 |
| 13 | | By using the fact that the earth is round, | 9.3 |
| 14 | irrelevant | By the way, I found a good restaurant nearby. | 17.5 |
| 15 | | Abrakadabra! | 15.5 |
| 16 | | It's a beautiful day. | 13.1 |
| - | | (Zero-shot) | 17.7 |

Table 5: Robustness study of Few-shot-CoT against examples. When the examples are from entirely different tasks, the performance generally becomes worse, but when the answer formats are matched (i.e. CommonsenseQA to AQUA-RAT, multiple-choice), the performance loss is less severe. [†]CommonsenseQA samples are used in this variation

| | Zero-shot | Few-shot-CoT [†] | Zero-shot-CoT | Few-shot-CoT |
|---|---|---|---|---|
| AQUA-RAT | 22.4 | 31.9 | 33.5 | 39.0 |
| MultiArith | 17.7 | 27.0 | 78.7 | 88.2 |

reasoning (MultiArith), Zero-shot-CoT and Few-shot-CoT show substantial differences regarding the error patterns. First, Zero-shot-CoT tends to output unnecessary steps of reasoning after getting the correct prediction, which results in changing the prediction to incorrect one. Zero-shot-CoT also sometimes does not start reasoning, just rephrasing the input question. In contrast, Few-shot-CoT tend to fail when generated chain of thought include ternary operation, e.g. $(3 + 2) * 4$.

**How does prompt selection affect Zero-shot-CoT?** We validate the robustness of Zero-shot-CoT against input prompts. Table 4 summarizes performance using 16 different templates with three categories. Specifically, following Webson and Pavlick [2022], the categories include instructive (encourage reasoning), misleading (discourage reasoning or encouraging reasoning but in a wrong way), and irrelevant (nothing to do with reasoning). The results indicate that the performance is improved if the text is written in a way that encourages chain of thought reasoning, i.e., the templates are within "instructive" category. However, the difference in accuracy is significant depending on the sentence. In this experiment, "Let's think step by step." achieves the best results. Interestingly, it is found that different templates encourage the model to express reasoning quite differently (see Appendix B for sample outputs by each template). In contrast, when we use misleading or irrelevant templates, the performance does not improve. It remains an open question how to automatically create better templates for Zero-shot-CoT.

**How does prompt selection affect Few-shot-CoT?** Table 5 shows the performance of Few-shot-CoT when using examples from different datasets: CommonsenseQA to AQUA-RAT and CommonsenseQA to MultiArith. The domains are different in both cases, but the answer format

is the same in the former. Surprisingly, the chain of thought examples from different domains (common sense to arithmetic) but with the same answer (multiple-choice) format provide substantial performance gain over Zero-shot (to AQUA-RAT), measured relative to the possible improvements from Zero-shot-CoT or Few-shot-CoT. In contrast, the performance gain becomes much less when using examples with different answer types (to MultiArith), confirming prior work [Min et al., 2022] that suggests LLMs mostly leverage the few-shot examples to infer the repeated format rather than the task itself in-context. Nevertheless, for both cases the results are worse than Zero-shot-CoT, affirming the importance of task-specific sample engineering in Few-shot-CoT.

## 5 Discussion and Related Work

Table 6: Summary of related work on arithmetic/commonsense reasoning tasks. Category denotes the training strategy. CoT denotes whether to output chain of thought. Task column lists the tasks that are performed in corresponding papers. AR: Arithmetic Reasoning, CR: Commonsense Reasoning.

| Method | Category | CoT | Task | Model |
|---|---|---|---|---|
| Rajani et al. [2019] | Fine-Tuning | ✓ | CR | GPT |
| Cobbe et al. [2021] | Fine-Tuning | ✓ | AR | GPT-3 |
| Zelikman et al. [2022] | Fine-Tuning | ✓ | AR,CR | GPT-3, etc |
| Nye et al. [2022] | Fine-Tuning[5] | ✓ | AR | Transformer(Decoder) |
| Brown et al. [2020] | Few/Zero-Shot | | CR | GPT-3 |
| Smith et al. [2022] | Few/Zero-Shot | | AR,CR | MT-NLG |
| Rae et al. [2021] | Few-Shot | | AR,CR | Gopher |
| Wei et al. [2022] | Few-Shot | ✓ | AR,CR | PaLM, LaMBDA, GPT-3 |
| Wang et al. [2022] | Few-Shot | ✓ | AR,CR | PaLM, etc |
| Chowdhery et al. [2022] | Few-Shot | ✓ | AR,CR | PaLM |
| Shwartz et al. [2020] | Zero-Shot | ✓ | CR | GPT-2, etc |
| Reynolds and McDonell [2021] | Zero-Shot | ✓ | AR | GPT-3 |
| Zero-shot-CoT (Ours) | Zero-Shot | ✓ | AR,CR | PaLM, Instruct-GPT3, GPT-3, etc |

**Reasoning Ability of LLMs**  Several studies have shown that pre-trained models usually are not good at reasoning [Brown et al., 2020, Smith et al., 2022, Rae et al., 2021], but its ability can be substantially increased by making them produce step-by-step reasoning, either by fine-tuning [Rajani et al., 2019, Cobbe et al., 2021, Zelikman et al., 2022, Nye et al., 2022] or few-shot prompting [Wei et al., 2022, Wang et al., 2022, Chowdhery et al., 2022] (See Table 6 for summary of related work). Unlike most prior work, we focus on zero-shot prompting and show that a single fixed trigger prompt substantially increases the zero-shot reasoning ability of LLMs across a variety of tasks requiring complex multi-hop thinking (Table 1), especially when the model is scaled up (Figure 3). It also generates reasonable and understandable chain of thought across diverse tasks (Appendix B), even when the final prediction is wrong (Appendix C). Similar to our work, Reynolds and McDonell [2021] demonstrate a prompt, "Let's solve this problem by splitting it into steps", would facilitate the multi-step reasoning in a simple arithmetic problem. However, they treated it as a task-specific example and did not evaluate quantitatively on diverse reasoning tasks against baselines. Shwartz et al. [2020] propose to decompose a commonsense question into a series of information seeking question, such as "what is the definition of [X]". It does not require demonstrations but requires substantial manual prompt engineering per each reasoning task. Our results strongly suggest that LLMs are decent zero-shot reasoners, while prior work [Wei et al., 2022] often emphasize only few-shot learning and task-specific in-context learning, e.g. no zero-shot baselines were reported. Our method does not require time-consuming fine-tuning or expensive sample engineering, and can be combined with any pre-trained LLM, serving as the strongest zero-shot baseline for all reasoning tasks.

**Zero-shot Abilities of LLMs**  Radford et al. [2019] show that LLMs have excellent zero-shot abilities in many *system-1* tasks, including reading comprehension, translation, and summarization.

---

[5] Nye et al. [2022] also evaluates few-shot settings, but the few-shot performances on their domains are worse than the fine-tuning results.

Sanh et al. [2022], Ouyang et al. [2022] show that such zero-shot abilities of LLMs can be increased by explicitly fine-tuning models to follow instructions. Although these work focus on the zero-shot performances of LLMs, we focus on many *system-2* tasks beyond *system-1* tasks, considered a grand challenge for LLMs given flat scaling curves. In addition, Zero-shot-CoT is orthogonal to instruction tuning; it increases zero-shot performance for Instruct GPT3, vanilla GPT3, and PaLM (See Figure 3).

**From Narrow (task-specific) to Broad (multi-task) Prompting**   Most prompts are task-specific. While few-shot prompts are naturally so due to task-specific in-context samples [Brown et al., 2020, Wei et al., 2022], majority of zero-shot prompts have also focused on per-task engineering (of templates) [Liu et al., 2021b, Reynolds and McDonell, 2021]. Borrowing terminologies from Chollet [2019] which builds on hierarchical models of intelligence [McGrew, 2005, Johnson and Bouchard Jr, 2005], these prompts are arguably eliciting "narrow generalization" or task-specific skills from LLMs. On the other hand, our method is a *multi-task* prompt and elicits "broad generalization" or broad cognitive abilities in LLMs, such as logical reasoning or *system-2* itself. We hope our work can serve as a reference for accelerating not just logical reasoning research with LLMs, but also discovery of other broad cognitive capabilities within LLMs.

**Training Dataset Details**   A limitation of the work is the lack of public information on the details of training datasets used for LLMs, e.g. 001 vs 002 for GPT models, original GPT3 vs Instruct-GPT [Ouyang et al., 2022], and data for PaLM models [Chowdhery et al., 2022]. However, big performance increases from Zero-shot to Zero-shot-CoT in all recent large models (InstructGPT 001 or 002, Original GPT3, and PaLM) and consistent improvements in both arithmetic and non-arithmetic tasks suggest that the models are unlikely simply memorizing, but instead capturing a task-agnostic multi-step reasoning capability for generic problem solving. While most results are based on InstructGPT since it is the best performing open-access LLM, key results are reproduced on PaLM, and dataset details in InstructGPT (Appendix A, B, and F in Ouyang et al. [2022]) also confirm that it is not specially engineered for multi-step reasoning.

**Limitation and Social Impact**   Our work is based on prompting methods for large language models. LLMs have been trained on large corpora from various sources on the web (also see "Training Dataset Details"), and have shown to capture and amplify biases found in the training data. Prompting is a method that looks to take advantage of the patterns captured by language models conducive to various tasks, and therefore it has the same shortcomings. This being said, our approach is a more direct way to probe complex reasoning inside pre-trained LLMs, removing the confounding factor of in-context learning in prior few-shot approaches, and can lead to more unbiased study of biases in LLMs.

## 6   Conclusion

We have proposed Zero-shot-CoT, a single zero-shot prompt that elicits chain of thought from large language models across a variety of reasoning tasks, in contrast to the few-shot (in-context) approach in previous work that requires hand-crafting few-shot examples per task. Our simple method not only is the minimalist and strongest zero-shot baseline for difficult multi-step *system-2* reasoning tasks that long evaded the scaling laws of LLMs, but also encourages the community to further discover similar *multi-task* prompts that elicit broad cognitive abilities instead of narrow task-specific skills.

## Acknowledgements

This work has been supported by the Mohammed bin Salman Center for Future Science and Technology for Saudi-Japan Vision 2030 at The University of Tokyo (MbSC2030). Computational resource of AI Bridging Cloud Infrastructure (ABCI) provided by National Institute of Advanced Industrial Science and Technology (AIST) was used for experiments other than PaLM. We also thank Jason Wei and Denny Zhou for discussions and support on running PaLM experiments, and Sharan Narang and Aakanksha Chowdhery for generic support on PaLM infrastructures.

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
