# OpenReview forum: "Large Language Models are Zero-Shot Reasoners"
_NeurIPS.cc/2022/Conference — NeurIPS 2022 Accept_

### Official Review · Reviewer_63Yb · 2022-07-10

**Rating:** 6
**Confidence:** 4
**Soundness:** 3 good
**Presentation:** 4 excellent
**Contribution:** 2 fair

**Summary:**

This is a paper that introduces a new zero-shot chain of thought prompting routine and examines its performance across various system2 tasks (e.g. coin flip, common sense, etc.) and on various models (e.g. gpt3, gpt-j, etc.). The prompt that works best in their experiments for zero-shot-cot is "Let's think step by step," which they use in most of their experiments. The new zero-shot-cot routine requires prompting the model twice, the second time just to extract the answer in the right structure. The experiment results show that zero-shot-cot (chain of thought) does better than zero-shot without chain of thought across most tasks. On MultiArith and GSM8K, zero-shot-cot does slightly worse than few-shot-cot methods but better than few-shot (without cot). The authors also show that zero-shot-cot performance scales well with model size and tests various types of zero-shot-cot prompts. (e.g. "Let's think...", "First")

**Questions:**

Questions:
1. One of the strengths of zero-shot-cot, the authors claim, is that it's task agnostic and doesn't require manual labeling. However, the "2nd prompt" aspect of the two-stage prompting process seems to require a manually engineered "answer trigger". Sure the labels are more confined (multiple choice, yes/no, etc.) but doesn't this make this method equally "manual"?
2. Are there cases where in the chain of reasoning, a wrong answer is derived, but the final answer (after 2nd prompt) is correct?
3. Why don't the authors make a graph of Table 3? Also I disagree that zero-shot/few-shot performance stays flat with model scaling. I think it also improves, just not as significantly.  (maybe phrase differently)
4. The examples used in the error analysis Table 4 are odd? and actually make the case the zero-shot-cot is doing good reasoning. Example 2's question in particular, "What would be necessary for getting in shape?" seems to obviously elicit (common-sensically) multiple answers, but the gold-label only picks one. Maybe these question/answer pairs here are not optimal.
5. How are the table 5 templates derived? I would like to know if these are the result of trial-and-error experiments or thought-out/reasoned examples.

Typos/writing:
1. line 32, p1 (or automatically becomes...) should be ( or automatically has become)
2. line 59, p2 (Figure 1) should (Table 2)
3. lines 105-112, p3: this paragraph could be written better to make clearer
4. lines 114, p3: [T] is supposed to be [Z]?
5. line 125, p4 (trigger depends on the answer format) should be (trigger depending on the answer format)


**Limitations:**

I enjoyed reading this paper and it's a good overview and survey of in-context zero-shot/few-shot learning. I would have appreciated this paper more if it were less about experiment results and more about an explanation as to why some things work better than others. I would like, for instance, if the authors could explain table 5 in more detail. Why do some templates work better than others? Or why zero-shot-cot doesn't perform better than zero-shot in commonsense reasoning tasks. Perhaps some other designed experiments could help fill this void. An idea would be to run an experiment where a whole bunch of templates are generated (like 100s) and tested to see which do best. I think an experiment like this would make the results look more objective and systematic and less random. I appreciate the paper as a good comparison of performance, but it feels lacking in its explanatory power.


**Strengths And Weaknesses:**


Originality:
This paper is somewhat original. The zero-shot-cot concept isn't novel but it was the first time various templates were experimented with in various set-ups. Lots of good experimental results here for future reference.

Quality:
This is a somewhat high quality paper with lots of experiments run on various models of different sizes. The authors cite all relevant works, to the best of my knowledge. Table 7 in particular is very good and shows the authors engaging with the in-context-learning literature.

Clarity:
Very high clarity and easy to read. Mostly well written with few errors, but Section 3 ("zero-shot chain of thought") and Section 4 (before 4.1) felt like they could be combined into one section.

Significance:
Somewhat significant results. Zero-shot-cot performance was interesting and somewhat surprising. But that it does worse than few-shot-cot was expected. Also I would like to know why zero-shot-cot doesn't improve over zero-shot on tasks like common sense reasoning.

---

> ### Author Response · Authors · 2022-08-02
> **Reply to Reviewer 63Yb**
>
> We thank all the reviewers for the valuable feedback. Please find our responses below.
> ***
> **About Question 1:** Surely, as Reviewer 39n8 has pointed out, 2nd prompt requires manual engineering, but with very limited efforts. So, we have modified Line 111 in the revised paper (“Zero-shot-CoT does not require such engineering” -> “Zero-shot-CoT requires less engineering”). Please be noted that we use the 2nd prompt mainly for quantitative evaluation for each task because manual evaluation (i.e., looking through reasoning sentences output by 1st prompt to check if the answer is correct or not) is impractical for experiments. We can skip the 2nd prompt if the situation allows so during the practical usage.
>
> **About Question 2:** We saw the case where a wrong chain of reasoning is derived, but the final answer (after 2nd prompt) is correct with a small percentage of a degree. For example, on MultiArith task, 6% of correct predictions encounter such a phenomenon for Zero-shot-CoT compared to 2% for Few-shot-CoT. On CommonSense task, a higher number of the ratio falls into this case because the task is a multiple choice task. See Appendix C for the details/examples within the revised paper.
>
> **About Question 3:** Following the reviewer’s suggestion, we have made a graph of the model scaling experiment with additional experiments on PaLM and moved the original table to Appendix E in the revised paper. Indeed, zero-shot/few-shot has slight performance gain as the model size becomes bigger. The finding is described in “Does model size matter for zero-shot reasoning?” at Section 4.1. (We have no intention to say “zero-shot/few-shot performance stays completely flat”, but instead “the performance does not increase or increases slowly as the model scale is increased, i.e., *the curve is mostly flat.*”)
>
> **About Question 4:** Please see Table 11 in Appendix B in the revised paper for samples in which zero-shot-cot is doing good/failed reasoning for each task. In addition, we have attached more samples about the error analysis at Table 21 in Appendix C. In addition, we have made all the experiment logs of Zero-shot-CoT for each task available at the following link. https://anonymous.4open.science/r/zero_shot_cot-1B2D/log/. We also made a small change of Table name from “Failure examples …” to “Examples generated by Zero-Shot-CoT on CommonsenseQA for Error Analysis.”
>
> **About Question 5:** Table 5 (in the original submitted paper) is the result of experiments after coming up with templates that we thought would encourage multi-step reasoning. The first idea was “Let’s think step by step”. The other templates were thought up later or picked up from related work later.
>
> **About Typos/writing:** Thank you for pointing these out. We have modified them in the revised paper.
>
> **About Limitations:** Currently, we cannot clearly answer why some templates work better than others, or why zero-shot-co t doesn't perform better than zero-shot in commonsense reasoning tasks. Following the suggestion by the reviewer, we have conducted extensive experiments with additional templates (but not a bunch of templates due to the lack of budget and time), in which we have clarified the case where prompting works well or poorly. We hope that the results help us keep on further research for understanding/solving the mechanism of prompting in the future work.
> ***
> Again, thank you for giving us the opportunity to strengthen our manuscript with your valuable comments and queries.

---

### Official Review · Reviewer_wzYQ · 2022-07-12

**Rating:** 6
**Confidence:** 4
**Soundness:** 4 excellent
**Presentation:** 4 excellent
**Contribution:** 3 good

**Summary:**

The paper shows that large language models can achieve better zero-shot performance on multiple reasoning tasks by simply attaching "Let's think step by step" at the end of prompts.

**Questions:**

Discussed in previous sections.

**Limitations:**

Yes, the paper does discuss its limitations.

**Strengths And Weaknesses:**

Strengths:
1. The proposed method is very simple and effective.
2. The paper is well written and presented.
3. Good analysis and interesting findings from the paper, such as: 1) model size matters for zero-shot learning, chain-of-thought works only when the model size is large enough; 2) zero-shot-cot generally works when the model is prompted with sentences encouraging reasoning, however, a few sentences significantly work better then the rest.

Weakness:
1.  The proposed method is more like a surprisingly effective hack/trick instead of a systematic technique. The whole paper boils down to one sentence "Let's think step by step". While the paper has well demonstrated the effectiveness of this trick, it can be strengthened by answering a few related questions: 1) how to find the optimal "template"? Is "let's think step by step" happen to work or the model really understands it? 2) What's the mechanism behind this? The template does not provide any extra information and why it can trigger the LLM?

2 The full pipeline in Figure 2 has two steps which I am not sure if it is necessary. I have tried a few examples using the Open AI play ground and it seems that by appending "Let's think step by step", the model naturally generates the final answer "375".

---

> ### Author Response · Authors · 2022-08-02
> **Reply to Reviewer wzYQ**
>
> We thank all the reviewers for the valuable feedback. Please find our responses below.
> ***
> **About Weakness 1:** How to find the optimal template and the mechanism of why “let's think step by step” trigger reasoning remains an open question. What we know currently is that some sentences encouraging reasoning work well as described at Table 4 in the revised paper. To clarify the case where proposed prompting works well or poorly, several new prompts were added for extensive experiments. Please see “Results of Additional experiment” thread for the details.
>
> **About Weakness 2:** The second prompt (for answer extraction) is introduced to make it easy to automatically extract the results in a proper format for evaluation. It is unnecessary if we use human evaluation. For example, the reasoning process does not necessarily output the answer as the desired format of  “yes/no” or “(A)(B)(C)(D)”. Please see Table 3 (in Main) and Table11 (in Appendix B) for the samples in the revised paper.
> ***
> Again, thank you for giving us the opportunity to strengthen our manuscript with your valuable comments and queries.

---

### Official Review · Reviewer_1ywu · 2022-07-13

**Rating:** 6
**Confidence:** 4
**Soundness:** 3 good
**Presentation:** 3 good
**Contribution:** 3 good

**Summary:**

The papers proposes to use chain-of-thought prompting on a large language model (LLM) in a zero-shot fashion to work on tasks that require multi-step logical reasoning or arithmetic. The key idea is to add a "let's think step by step" at the end of the prompt to elicit a step-by-step approach from an LLM to do these tasks. The framework is two-stage, a first step a detail explanation is extracted out by the adding "let's think step by step" and in the second step, an output-specific answer prompt is used to get the answer in the required format for the task. The authors compare this prompt with normal manual zero-shot prompting, few-shot chain-of-thought prompting and manual few-shot prompting baselines on 12 datasets across 13 models in total. The zero-shot chain-of-thought prompting outperforms the zero-shot manual prompting in all experiments except the commonsense reasoning tasks, works better than few-shot manual prompting, but underperforms few-shot chain-of-thought prompting.

**Questions:**

- I believe the title of the paper should be more specific. The experiments in the paper itself don't really end up in the claim of the title without proper data overlap and other analysis.
- For instructGPT, is it even clear what data it was fine-tuned on? If not, then it is not true zero-shot right?

**Limitations:**

Yes

**Strengths And Weaknesses:**

- The paper provides another verification in the debate of prompting that LLMs have learnt a vast amount of knowledge and certain prompts can elicit specific information from these LLMs.
- The method in itself is simple and it is interesting that it works so well across the tasks and models.
- The authors conduct various experiments to see what matters for the performance including testing out various other prompts that are similar in semantic sense.
- The method is effective when the model size is large, but can be detrimental when model size is small.
- The amount of difference that we are seeing in performance with different prompts worries me as we are basically moving more and more towards prompt-tuning where as before we were doing hyper-parameter tuning. The paper doesn't do any sort of leakage analysis which is important because in this we might just be distilling a specific part of the pretraining dataset. Overall, the findings in this paper are probably an artifact of the pretraining data and since this paper highlights that I believe it is an important contribution.

---

> ### Author Response · Authors · 2022-08-02
> **Reply to Reviewer 1ywu**
>
> We thank all the reviewers for the valuable feedback. Please find our responses below.
> ***
> **About Question 2:** For Instruct-GPT3, fine-tuned data(question part) was collected from users via OpenAI API, the output(answer part) was written by human labellers[*a]. The dataset itself is not published, but some samples (See “Illustrative user prompts” at Appendix A and “Labeler demonstration” at Appendix F) in [*a] confirm that it is not specially engineered for multi-step reasoning. “Zero-shot” approach generally refers to the fact that the model does not take any samples as in-context inputs in contrast to the few-shot approach, so our method still remains “Zero-shot”. The point is that our method can elicit multi-step reasoning ability which was not obvious for the normal zero-shot approach. The same scale of improvement is observed from additional experiments by PaLM(540B) model [*b], which does not involve instruction finetuning. Please see “Results of Additional experiment” thread for the details about experiments with PaLM.
>
> **About Question 1:** Given our reply to Question2 (above), we still believe that the current title is appropriate. However, we welcome any suggestion about the alternatives of the title.
> ***
> Again, thank you for giving us the opportunity to strengthen our manuscript with your valuable comments and queries.
>
> Reference:
>
> [*a] Ouyang, Wu et al. “Training language models to follow instructions with human feedback” (2022)
> [*b] Tay, Dehghani et al. “PaLM: Scaling Language Modeling with Pathways” (2022)

---

> > ### Comment · Reviewer_1ywu · 2022-08-09
> > **Thanks for the rebuttal**
> >
> > I thank the authors for the rebuttal. About the title, I am still not convinced and it can be more specific, something like: "Chain-of-thought reasoning with LLMs using specific prompt". Also, I don't believe this is actually zero-shot in all cases because of the tuning that had to be done to reach this prompt (see [this](https://proceedings.neurips.cc/paper/2021/hash/5c04925674920eb58467fb52ce4ef728-Abstract.html)). Given that BLOOM and other open models are recently released out there with knowledge about pretraining data, the distillation experiment can now be conducted properly on whether the task data is somehow part of the pretraining data. Given all of these concerns, I would like to keep my rating as is.

---

> > > ### Author Response · Authors · 2022-08-09
> > > **Thank you for the further comments**
> > >
> > > We thank the reviewer for reading our rebuttal and further comments.
> > > ***
> > > - Regarding the comments "whether the task data is somehow part of the pretraining data", please see the relevant part of the rebuttal to Reviewer zon4, which is reprinted at the bottom in this thread.
> > > - In our paper, we use the terms “zero-shot”, “few-shot” to just indicate the number of demonstrations provided at test time ([GPT-3 literature](https://proceedings.neurips.cc/paper/2020/hash/1457c0d6bfcb4967418bfb8ac142f64a-Abstract.html)). Based on this, our title uses “zero-shot” to contrast with the existing “few-shot” based approaches (e.g. [URL](https://arxiv.org/abs/2201.11903)).
> > > - There are indeed several open-source large language models (OPT, Bloom, GLM, etc.), but most of which are released after the NeurIPS submission deadline.
> > > ***
> > > **Excerpt from the rebuttal for Reviewer zon4** "Yes, the knowledge definitely comes from pre-training corpus. However, the fact that our method works only at scale alludes to some potential other phenomenon as discussed in [*c]. Specifically, this paper shows the emergence of “step-by-step reasoning ability” even without few-shot examples. Moreover, it does not just memorize the reasoning examples, but we show that it generalizes the reasoning ability to new examples (tasks). To validate that, we used the various well established benchmarks suitable for multi-step reasoning tasks[*d]. It is worth noting that most of the tasks are not included in (pre-)training corpus with a high probability. For example, Instruct-GPT3 (text-davinci-002) uses data up to June/2021 for training [*e], while “Coin Flip”[*c] task is released after that. GSM8K[*g] was created by “human problem writers” and published at the end of 2021, which means the pre-trained dataset does not include it. We further confirmed with authors of GPT-3, Instruct-GPT3, PaLM that datasets such as GSM8K are not included in training (which if so would invalidate most of the recent NLP papers on those benchmarks)."
> > > ***

---

### Official Review · Reviewer_39n8 · 2022-07-15

**Rating:** 6
**Confidence:** 3
**Soundness:** 3 good
**Presentation:** 3 good
**Contribution:** 3 good

**Summary:**

This paper explores the zero-shot reasoning abilities of large language models. Extending from recent work on few-shot chain-of-thought prompting, in this work the authors propose to elicit step-by-step reasoning by simply adding “let’s think step by step” in the beginning of the input, then adopt a two-stage prompting approach (Figure 2). Results on a collection of 12 reasoning tasks suggest that zero-shot chain-of-thought (this work) greatly outperforms the raw zero-shot baseline and outperforms 8-shot few-shot (in-context) baseline. Further analysis is conducted on model size, prompt selection etc.

**Questions:**

I think the paper is very informative already but I have several questions out of curiosity.
* Is it possible that the (pre-)training corpus contain some step-by-step reasoning examples that leads to the zero-shot reasoning behavior? Does any of the prompt appear in the training corpus already? Would like to hear the authors comments on this.
* In Table 5, would be helpful to try more diverse phrasing of the prompts and perhaps 1 or 2 “bad” prompts (similar to the analysis in Fig. 5 in https://arxiv.org/abs/2109.01247).

Minor suggestion:
* I believe there is still a little “prompt engineering” efforts needed for zero-shot CoT as described in Line 125. Perhaps the claim in Line 111 is a little strong.

**Limitations:**

Limitations are discussed in Section 5

**Strengths And Weaknesses:**

Strengths
* Comprehensive experiments to demonstrate that prompting with “let’s think step by step” greatly improves zero-shot capabilities of large language models on reasoning tasks. The method is simple yet effective.
* The authors also provide a detailed survey of relevant zero-shot/few-shot approaches in Sec 5 and Table 7, which is a useful roadmap for researchers interested in the topic.
* The examples in the appendix are enjoyable to read. Sample Study section is the hidden gem in the appendix.
* Clear writing and visualization.

Weaknesses/Limitations
* According to Table 3, the proposed method is only effective with 175B Original-GPT3 or Instruct-GPT3 model. The practical utility of the method is limited. It would also be great to have the cost of using GPT-3 listed in the paper for reference.
* As prompting is a new and rising field, I’m not certain whether the major finding in this paper (“let’s think step by step”) serve as solid contribution for NeurIPS publication. I hope the authors can further convince me on the impact, utility or significance of their findings.

---

> ### Author Response · Authors · 2022-08-02
> **Reply to Reviewer 39n8**
>
> We thank all the reviewers for the valuable feedback. Please find our responses below.
> ***
> **About Weakness 1:** To validate that our proposal is effective on not only GPT-3s, we have conducted additional experiments on PaLM(540B)[*a]. Please see “Results of Additional experiment” thread for the experiment details. Rather than practical usage, our work focuses on showing that certain prompts can elicit the emergent behaviors without explicitly specifying a range of problems/context. The approximate estimated cost of zero-shot-cot using GPT-3(in the case of davinci, text-davinci-001, text-davinci-002) for some tasks is listed below. Please note that GPT-3 is a token based-billing, so the cost varies depending on reasoning length [*b]. The list below is based on the assumption that reasoning length reaches 128 (maximum length).
> | |  | SingleEq | AddSub | MultiArith | AQUA-RAT | SVAMP | GSM8K|
> | ---- | ---- | ---- |---- | ---- | ---- | ---- |---- |
> |1 | Avg. # of words in one question | 27.4 | 32.5 | 33.8 | 51.9 | 31.8 | 46.9|
> |2 | Avg. # of estimated words for reasoning | 128.0 | 128.0 | 128.0 | 128.0 | 128.0 | 128.0|
> |3 | Avg. # of estimated words for one sample (Notes: run twice for 2nd prompts) | 310.8 | 321.0 | 323.6 | 359.8 | 319.6 | 349.8|
> |4 | Avg. # of estimated tokens for one sample (Notes: 1 word ≒ 4/3 tokens) | 414.4 | 428.0 | 431.5 | 479.7 | 426.1 | 466.4|
> |5 | TTL # of samples in dataset | 508.0 | 395.0 | 600.0 | 254.0 | 1,000.0 | 1,319.0|
> |6 | TTL # of estimated tokens for dataset (Notes: No.4 * No.5) | 210,515.2 | 169,060.0 | 258,880.0 | 121,852.3 | 426,133.3 | 615,181.6|
> |7 | Billing rate per token (US$) | 0.00006 | 0.00006 | 0.00006 | 0.00006 | 0.00006 | 0.00006|
> |8 | Estimated Billing for dataset (US$) (Notes: No.6 * No.7) | **12.6** | **10.1** | **15.5** | **7.3** | **25.6** | **36.9** |
>
> **About Weakness 2:** Our approach does not just provide incremental performance gain, but also elicit a new behavior (Chain-of-Thought: multi-step thinking/reasoning) from large language models by a single prompting for a wide variety of tasks. This finding is novel and it opens up the question of connecting emergent properties, compositional generalization and scale from the pre-training corpora.
>
> **About Question 1:** It may occur that (pre-)training corpus already contains some step-by-step reasoning examples as well as the corresponding prompt, but we have not been able to isolate this. What we do know is that Let’s think step by step never appears in C4, and the few appearances of step by step largely do not appear in reasoning/CoT style setups.
>
> **About Question 2:** To clarify the case where prompting works well or poorly, several new prompts were added for extensive experiments. Please see “Results of Additional experiment” thread for the experiment details.
>
> **About Minor suggestion:** Thank you for pointing this out. We have modified it in the revised paper. (“Zero-shot-CoT does not require such engineering” -> “Zero-shot-CoT requires less engineering”)
> ***
> Again, thank you for giving us the opportunity to strengthen our manuscript with your valuable comments and queries.
>
> Reference:
>
> [*a] Tay, Dehghani et al. “PaLM: Scaling Language Modeling with Pathways” (2022)
> [*b] https://openai.com/api/pricing/

---

### Official Review · Reviewer_zon4 · 2022-07-16

**Rating:** 5
**Confidence:** 4
**Soundness:** 3 good
**Presentation:** 2 fair
**Contribution:** 2 fair

**Summary:**

The authors propose a two-stage template-based prompt method for general NLP tasks, named zero-shot-CoT. In this work, it proves that large pre-trained language models have a talent to extract reasoning chains without any few-shot examples for finetuning or imitating. They also investigate the inference ability of LMs with different scales on various datasets.

**Questions:**

1. The authors declare that larger LMs have better ability of zero-shot reasoning, but they only provide results of GPT-3 models with different scales except T0. Can this conclusion apply to models with other structures? Will there be a upheaval as the scale increasing?
2. Larger LMs can memorize more knowledge from pre-training corpus, therefore, these remarkable experimental results of large LMs may come from what they have learnt from pre-training. Will the proposed method be effective on more tasks of some specific domains like commonsense, news QA, or even bioscience?
3. A trivial error: In P3-L114, `Q:[X] A:[Z]` should be `Q:[X] A:[T]`?


**Limitations:**

The authors address the limitations.

**Strengths And Weaknesses:**

Strengths:
1. In this two-stage framework (called CoT), the first step extracts reasoning chains and the second can summary the final answers from extracted chains.
2. The authors also employ post-processing strategies to acquire clean answers.

Weakness:
1. There exists many zero-shot prompt methods by adding a short prompt like `Let’s think step by step`, which means this work is not novel enough nowadays.
2. Large language models have been proven to have satisfying potential of zero-shot reasoning, especially GPT-3 and its variants.
3. Although there are some analyses about the robustness of different prompts in table 5 and 6, it is not enough to prove the proposed method can solve general tasks. The authors should provide detailed results on every tasks but not a score on a specific task.

---

> ### Author Response · Authors · 2022-08-02
> **Reply to Reviewer zon4**
>
> We thank all the reviewers for the valuable feedback. Please find our responses below.
> ***
> **About Weakness 1:** We agree that the zero-shot prompting itself method is a well-known approach. However, as discussed in the section 5 of the submitted manuscript, most prior works focus on system-1 tasks that do not require multi-step reasoning. Our work is the first study that shows that a single zero-shot prompt (Let’s think step by step) can elicit a new behavior (chain-of-thought: multi-step thinking/reasoning), and it significantly jumps up the performances on many reasoning tasks that require multi-step thinking without changing the prompt. This “broad generalization of a single prompt” finding is novel and has potential to make a new paradigm of reasoning by large language models.
>
> **About Weakness 2:** To validate that our proposal is effective on not only GPT-3s, we have conducted additional experiments on PaLM(540B)[*a]. Please see “Results of Additional experiment” thread for the experiment details. Additionally, it would be helpful if the reviewer points us to specific papers with thorough experimentations that showcase zero-shot reasoning capability (e.g. zero-shot wasn’t even considered in CoT paper [*d], as the main NLP community did not believe such a complex behavior could emerge without task-specific carefully-crafted few-shot examples).
>
> **About Weakness 3:** As for the generalizability of our method, Table 1 describes well that our method can be applied to a wide variety of tasks. The detailed results can be found at Appendix B. In addition, we have made all the experiment logs of Zero-shot-CoT for each task available at the following link. https://anonymous.4open.science/r/zero_shot_cot-1B2D/log/. We apologize if our understanding of your point is incorrect.
>
> **About Question 1:** The same reply as Weakness 2. If the question is about the structure of the model, [*b] insists that the choice of the denoiser (how to mask the pretraining corpus) has significantly more impact than the architecture (encoder-decoder v.s. decoder-only). In addition, to show that larger LMs have better ability of zero-shot reasoning for different models other than GPT-3, we have also conducted model scale study with PaLM. Please see “Results of Additional experiment” thread for the experiment details.
>
> **About Question 2:** Yes, the knowledge definitely comes from pre-training corpus. However, the fact that our method works only at scale alludes to some potential other phenomenon as discussed in [*c]. Specifically, this paper shows the emergence of “step-by-step reasoning ability” even without few-shot examples. Moreover, it does not just memorize the reasoning examples, but we show that it generalizes the reasoning ability to new examples (tasks). To validate that, we used the various well established benchmarks suitable for multi-step reasoning tasks[*d]. It is worth noting that most of the tasks are not included in (pre-)training corpus with a high probability. For example, Instruct-GPT3 (text-davinci-002) uses data up to June/2021 for training [*e], while “Coin Flip”[*c] task is released after that. GSM8K[*g] was created by “human problem writers” and published at the end of 2021, which means the pre-trained dataset does not include it. We further confirmed with authors of GPT-3, Instruct-GPT3, PaLM that datasets such as GSM8K are not included in training (which if so would invalidate most of the recent NLP papers on those benchmarks).
>
> **About Question 3:** This is a typo. Thank you for pointing this out. We have modified it in the revised paper.
> ***
> Again, thank you for giving us the opportunity to strengthen our manuscript with your valuable comments and queries.
>
> References
>
> [*a] Tay, Dehghani et al. “PaLM: Scaling Language Modeling with Pathways” (2022)
> [*b] Chowdhery, Narang et al. “Unifying Language Learning Paradigms” (2022)
> [*c] Wei, Tay et al. “Emergent Abilities of Large Language Models” (2022)
> [*d] Wei, Wang et al. “Chain of Thought Prompting Elicits Reasoning in Large Language Models” (2022)
> [*e] https://beta.openai.com/docs/models/gpt-3
> [*f] https://github.com/google/BIG-bench
> [*g] Cobbe, Kosaraju et al. “Training Verifiers to Solve Math Word Problems” (2021)

---

### Author Response · Authors · 2022-08-02
**Results of Additional Experiments**

Following reviewers' comments, we have conducted the following additional experiments and included them in the revised paper.
***
**1. Additional experiments on PaLM 540B**
To validate that our proposal is effective on not only (Instruct)GPT-3s, we have conducted additional experiments on PaLM 540B[*a].

**1-1. Zero-shot-CoT with PaLM 540B and comparison with other methods**
It is confirmed that Zero-Shot-CoT also works on PaLM 540B, bringing GSM8K performance from 12.5% to 43.0%, and MultiArith from 25.5% to 66.1%. This shows that our method works on large pretrained models other than GPT-3, and does not require instruction-finetuning or specially-curated data [*d]. ***These results (below) are included at Table 2 in the revised paper.***

|    |  MultiArith  |  GSM8K  |
| ---- | ---- | ---- |
|  PaLM 540B: Zero-Shot  |  25.5  |  12.5  |
|  **PaLM 540B: Zero-Shot-CoT**  |  **66.1**  | **43.0**  |
|  **PaLM 540B: Zero-Shot-CoT + self consistency**  |  **89.0**  |  **70.1**  |
|  PaLM 540B: Few-Shot  |  -  |  17.9  |
|  PaLM 540B: Few-Shot-CoT  |  -  |  56.9  |
|  PaLM 540B: Few-Shot-CoT + self consistency  |  -  |  74.4  |

***Further experiment results (below) on other tasks and with self-consistency are described in Appendix D in the revised paper.*** We have confirmed that the Zero-shot-CoT performance further boosts if self consistency[*c](generating multiple reasoning paths by random sampling and majority voting of predictions) is applied, indicating the potential for a wide variety of improvements and applications.

|    | AQUA-RAT  |  SVAMP  | GSM8K  | MultiArith  |
| ---- | ---- | ---- |---- | ---- |
|  Zero-Shot  |  23.4  | **63.1**  | 12.5  | 25.5  |
|  Zero-Shot-CoT  |  **36.1**  | **63.1**  | **43.0**  | **66.1**  |
|  Zero-Shot-CoT + self consistency  | **46.5**  | **80.5**  | **70.1**  | **89.0**  |
|  Few-Shot-CoT  | 35.8 | 79.0 | 56.9 | -  |
|  Few-Shot-CoT + self consistency  | 48.3  | 86.6  | 74.4  | -  |



**1-2. Model scale study with PaLM (8B, 62B, and 540B)**
The result with PaLM demonstrates that Zero-shot-CoT performance improves non-linearly as the model size is large enough, while Zero-shot performance increases just linearly. This indicates that the emergent ability[*e] by zero-shot chain of thought reasoning exists not only on GPT-3s but also on PaLM. Please note that ***the original table + the following results were moved to Appendix E in the revised paper. Instead, Figure 3 is newly created in Main to visualize the curves of model scaling for GPT-3 and PaLM.***

|  PaLM on GSM8K  | 8B  | 62B | 540B |
| ---- | ---- | ---- |---- |
|  Zero-Shot  | 2.1  | 7.0  | 12.5  |
|  Zero-Shot-CoT  | **2.4**  | **10.5**  | **43.0**  |


***
**2. Additional experiments for template study**
To clarify the case where prompting works well or poorly, following [*b], ***several new prompts (No.10-16: See below) were added to Table 4 in the revised paper*** for extensive experiments and all templates are classified into three categories: instructive (encourage reasoning), misleading (discourage reasoning or encouraging reasoning but in a wrong way), and irrelevant (nothing to do with reasoning). The results indicate that the performance is improved if the text is written in a way that encourages chain of thought reasoning, i.e., the templates are within the "instructive" category.  In contrast, when we use misleading or irrelevant templates, the performance does not improve. It remains an open question how to automatically create better templates for Zero-shot-CoT. ***Some generated samples by the new templates are added at Table 12 in Appendix B in the revised paper.***

| No. | Category | Template | Accuracy |
| ---- | ---- | ---- |---- |
| 1 | instructive | Let’s think step by step.  | 78.7 |
| … |  | … | … |
| 10 | misleading | Don't think. Just feel. | 18.8 |
| 11 |  | Let's think step by step but reach an incorrect answer. | 18.7 |
| 12 |  | Let's count the number of "a" in the question. | 16.7 |
| 13 |  | By using the fact that the earth is round, | 9.3 |
| 14 | irrelevant | By the way, I found a good restaurant nearby. | 17.5 |
| 15 |  | Abrakadabra! | 15.5 |
| 16 |  | It's a beautiful day. | 13.1 |
| - | - | (Zero-shot) | 17.7 |


***
References

[*a] Tay, Dehghani et al. “PaLM: Scaling Language Modeling with Pathways” (2022)
[*b] Webson, Pavlick "Do Prompt-Based Models Really Understand the Meaning of their Prompts?" (2022)
[*c] Wang, Wei et al. “Self-Consistency Improves Chain of Thought Reasoning in Language Models” (2022)
[*d] Ouyang, Wu et al. “Training language models to follow instructions with human feedback” (2022)
[*e] Wei, Tay et al. “Emergent Abilities of Large Language Models” (2022)

---

### Author Response · Authors · 2022-08-10
**A Gentle Reminder**

**To : Area Chairs and all the reviewers**

As of now, we have got only one response from one reviewer out of 5.
If any of the other reviewers have any question / comment about the the rebuttal or the results of the additional experiments, we would be happy to answer them.

We are very sorry for the last minute reminder during your busy time.

---

### Meta-Review · Area_Chair_a65m · 2022-08-21

**Recommendation:** Accept
**Confidence:** Certain

**Metareview:**

This paper shows that large language models, if prompted in a particular way, can drastically improve their performance on zero-shot reasoning tasks. Specifically, simply adding "Let's think step by step" to the prompt causes accuracy to improve by a huge margin. The paper provides a wide variety of experimental results corroborating the central claim, and situates itself well in the existing literature.

The reviewers are on the fence about this work, and as an area chair I find myself similarly torn: the contribution itself is unequivocally mundane -- and I mean this without any offense to the authors, but as a matter of principle I do not think that "prompt engineering", much like "tuning on the test set", is a scientific discipline that deserves much (if any) attention in a machine learning conference. That said, this paper has two main things speaking for it: the fact that **the improvement is so large** AND the fact that **it works increasingly well with scale**. Oftentimes simple ideas are the most impactful ones, and I think that this work can have an impact on the field as a sort of existence proof: the mere observation that a single prompt can lead to such large gains, and that these gains only come about in very large-scale models, either points to serious flaws in our evaluation capabilities (from data overlap to faulty benchmarks) or some as-yet-poorly-understood capability in these models. This work opens up the way to important follow-up work that examines these questions more closely and conclusively, and consequently I think the paper in its current form merits acceptance.

The main shortcoming of the initial submission was that it only used one model, but that has been addressed in the rebuttal with the addition of PaLM. I would strongly encourage the authors to heed Reviewer 1ywu's advice to try to add more models and look more closely at the training data. While it is true that open source models like Bloom (where the training data is publicly available) were not available yet by the time of submission, they are now, and the paper would be considerably strengthened by an analysis into the causes of this effect. In addition, I was a bit disappointed that the "instructive" prompts were not very diverse -- it is well known that large language models are particularly susceptible to minor variations in prompts, so it would have been interesting to try a more complete spectrum of prompts including things like "Let’s solve the problem bit by bit", "If we solve the problem stepwise/gradually/piecemeal", "Let’s be smart about this", "Let’s be careful/make sure we get the right answer" to try to get more clarity on what exactly is happening (and to show the audience they need to be careful with overly trusting their prompts will do the right thing). Lastly, the work almost invites anthropomorphization, along the lines of "humans also do better when prompted to solve a problem step by step", and I would have liked to see a more serious discussion of this aspect of the work (and hence why I think further analysis of training data/prompt variations should be included in the final version).

**Award:**

No

---

### Decision · Program_Chairs · 2022-09-14

Accept